# Opportunity for GNSS Reflectometry in Sensing the Regional Climate and Soil Moisture Instabilities in Myanmar

**Aung Lwin** [1,2,*] [ID], **Dongkai Yang** [1], **Xuebao Hong** [1], **Bo Zhang** [1], **Baoyin Zhang** [1] and **Cheraghi Shamsabadi Sara** [1]

1   School of Electronic and Information Engineering, Beihang University, Beijing 100191, China;
    edkyang@buaa.edu.cn (D.Y.); joyce_hong2008@buaa.edu.cn (X.H.); bozhang@buaa.edu.cn (B.Z.);
    zbyun2020@buaa.edu.cn (B.Z.); saracheraghi@buaa.edu.cn (C.S.S.)
2   Remote Sensing and GIS Research Centre, Yangon Technological University, Yangon 11101, Myanmar
*   Correspondence: aunglwin@buaa.edu.cn; Tel.: +86-959-7988-14140

**Abstract:** The climate crisis is happening globally, and the consequent process has revealed soil evolution and meteorological interactions. The GNSS reflectometry (GNSS-R) technique recently encompassed sea surface monitoring, land changes, and snow sensing in addition to position, navigation, and timing. After the launch of NASA's eight CYGNSS satellites, spaceborne soil moisture retrieval has become more opportune in a global and regional investigation. The research carried out by the CYGNSS DDM SNR with SMAP data to correlate diurnal mean soil moisture sensing was analyzed in the regional study of Myanmar, which is prone to climatic and weather conditions. The results showed that spaceborne GNSS-R soil moisture sensitivity was very useful during seasonal changes in regional observation. The DDM SNR surface reflectivity was strongly correlated with soil moisture according to surface temperature variations prepared from SMAP passive reflectometry. Sentinel SAR-1 data included the validation and verification of flood-prone areas affected by tropical storm surges or weather depressions in the monsoon season. The availability of surface reflectivity primarily relied on the surface roughness, surface temperature, and vegetation opacity for soil moisture retrieval.

**Keywords:** GNSS-R; DDM; CYGNSS; SMAP; Sentinel SAR-1; soil moisture; geophysical parameters

## 1. Introduction

The GNSS technology's capability in positioning, navigation, and timing information is extracted from direct signals. In 1993, Martin Neira initiated the first bistatic radar remote sensing concept, the Passive Reflectometry and Interferometry System (PARIS) [1]. This system relies on the L-band of the Global Positioning System (GPS) signal for an ocean altimetry application. Ground, air, and spaceborne observations triggered by GNSS-R have progressed to detection of sea surface anomalies, snow, and land characteristics.

GNSS-R observatory satellites such as TDS-1 and the CYGNSS mission primarily serve sea surface observation purposes. By simultaneously receiving reflected signals, they can sense soil moisture, vegetation, snow, and environmental characteristics. After the launch of the eight CYGNSS satellites, the GNSS-R mission extended to retrieving soil moisture contents from the tropical regions. Despite the advances, spaceborne soil moisture sensing is still uncertain and preliminary because of the data limitation, collocation, and footprint heterogeneity [2]. A geophysical parameter such as surface roughness is challenging to define when dealing with coherent or incoherent surface, and it remains challenging to retrieve those uncertain parameters from spaceborne soil moisture observations [3].

Two main GNSS-R satellites, TDS-1 (2014) and CYGNSS (2016), have provided scientific observations of features such as ocean and sea surface anomalies, soil moisture and vegetation, wetlands inundation characterization, and the dynamics of hurricane- and tsunami-driven flooding [4–9]. In 2015, the Soil Moisture Active Passive (SMAP) satellite

radar receiver switched to 1227.45 MHz, enabling GPS L2 signals, and became one of the first GNSS reflectometry missions collecting GPS bistatic radar measurements [10].

Soil moisture observation occurs in the microwave region of the electromagnetic spectrum. Soil moisture primarily affects the dielectric constant of the soil medium. Therefore, the reflectivity or emissivity of the surface must be detected. To reduce atmospheric attenuation and to detect more penetrating vegetation with a longer wavelength, microwave frequencies in the 1–3 GHz range are ideally relevant for observing soil moisture [11]. Microwave sensors measure soil moisture, but new spaceborne GNSS observation is still being developed [12].

A GNSS-R receiver using a bistatic radar signal is comparable to passive radiometer soil moisture measurements because of the surface dielectric properties and surface roughness conditions [13]. Spaceborne GNSS-R has a potentially more acceptable spatial resolution than microwave radiometry because of the highly stable carrier and code modulations with delayed Doppler mapping capabilities. However, these actual measurement sensitivities to bio-geophysical variables of interest, such as soil moisture content and vegetation biomass, have not been assessed conclusively [14]. Chew et al. (2020) concluded that all require soil moisture data on short time scales, and a daily soil moisture reading may be able to provide complete information on soil moisture dynamics at needed time scales [15].

Drought is a disaster event and is primarily related to the consequences of soil moisture and evapotranspiration (ET). Gavahi et al. (2020) evaluated soil moisture predictions by multivariate data assimilation of ET and SM compared to the univariate method. They found that ET and SM contributed more to improving drought monitoring than any univariate assimilation configuration [16]. Humphrey et al. (2021) pointed out that soil moisture–atmosphere feedback indirectly amplifies temperature and humidity anomalies and enhances the direct effects of soil water stress. They found that most of the global variability in modeled land carbon uptake is driven by temperature and vapor pressure deficit effects that are controlled by soil moisture [17].

The L-band GNSS-R receiver detecting the reflected signal from the Earth's surface can simultaneously retrieve coherent and incoherent scattering, relying on the surface roughness phenomenon. However, a proportion of the incoherent component was directly ignored in previous research with GNSS-R soil moisture content retrieval owing to its relatively low significance [18]. Soil moisture retrieval is quite mature and has largely been investigated by ground and air, and is still applicable from space, whereas for biomass and freeze/thaw, the project will be more explorative in future missions. Soil moisture, vegetation biomass, and freeze/thaw processes play an essential role in studying and understanding the carbon and energy cycle of the Earth and the monitoring of the environment [19].

The majority of the previous GNSS-R-based SM retrievals also created their models and enhanced their performances with spaceborne data or point scale in situ observation. However, this is not adequate, and each research project adopted critically unrelated inversion methods owing to differences in (1) assumptions regarding gridding, open water masking, and surface conditions, (2) ancillary data requirements, (3) validation and reference datasets, (4) time spans, (5) models, and (6) spatial coverage [20].

This study aims to understand the basic concept of GNSS-R to improve the accuracy and discover the optimal solution for the extraction of Earth's surface characteristics by relying on space-based GNSS-R data opportunities. This research highlights achieving a daily 9-km gridded SM reading within the CYGNSS spatial coverage area ($\pm 38°$ latitudes) and formulating the predictions against the Soil Moisture Active Passive (SMAP) mission-enhanced SM L3 readings of 9 km $\times$ 9 km spatial resolution for soil moisture retrieving. The spaceborne GNSS-R contribution to the regional study revealed soil moisture sensitivity, land stability, and influence factors such as surface temperature, roughness, and vegetation opacity to the soil moisture analysis.

## 2. Physiography and Climatic Condition of Myanmar

### 2.1. Surface Roughness

Myanmar is bordered on the west and northwest by Bangladesh and India, and on the northeast and east (the Golden Triangle region) by China, Laos, and Thailand. The principal drainage systems are the extensive Ayeyarwady-Chindwin-Sitaung system, which drains the west and north regions, and the Thanlwin River system, originating in Tibet, drains the east part of the country. The topographic features of Myanmar become lower from north to south, from mountains and plateaus in the north to plains in the south. As shown in Figure 1, the main geographical features can be categorized into highlands in the east, mountains and coastal areas in the west, and plains in the central basin.

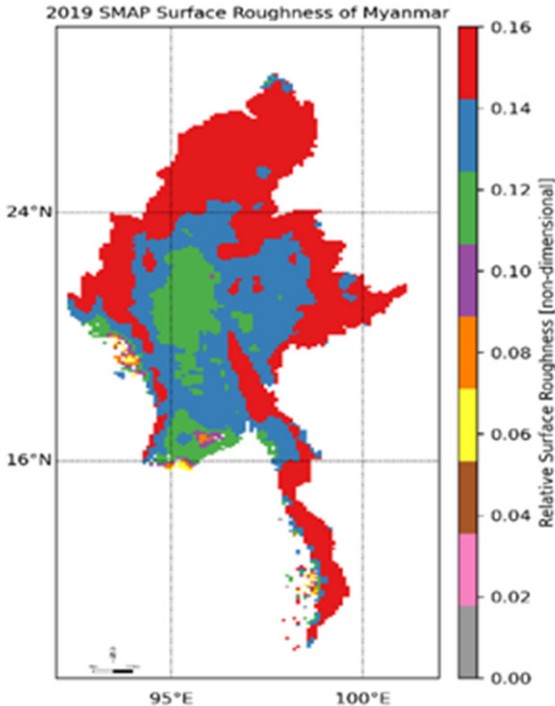

**Figure 1.** Surface roughness map prepared from the SMAP L3 Passive Radiometer.

Surface roughness, which is the h parameter, closely influences soil moisture availability. SMAP radiometer sensors operate in the microwave L-band, and the effect of surface roughness is a key parameter compared to vegetation. In theory, surface roughness is less sensitive than vegetation biomass in microwave emission and backscatter to soil moisture variations. The vegetation canopy and seasonal weather conditions are also significant factors in soil moisture sensing.

In the CYGNSS theoretical prediction of coherent reflectivity as a function of elevation angle, rough topography contributed to reducing the spreading of the DDM simultaneously; surface roughness increases the spreading with more significant roughness because more specular reflection points contribute to the total scattered field [10]. If the GNSS bistatic radar receivers can quantify dielectric impacts and surface roughness more precisely, the spaceborne GNSS-R technique will be a promising way to retrieve soil moisture at relatively better spatial and temporal resolution.

### 2.2. Climatic Conditions

Myanmar is the largest country in the Mekong region and the second-largest country in Southeast Asia. Myanmar has three seasons, with a tropical monsoon, hot, wet, and cold climate every four months. In recent decades, the weather and environmental crisis of Myanmar has increased remarkably, resulting in soil erosion and fertility loss, shortage in

water availability, decreased river flows, and inland flooding and storm surges, which are significant disaster impacts.

In Myanmar, we observed the effects of climate change, which are increasingly inconsistent rainfall patterns, higher temperatures that reduce agricultural productivity in the central dry zone, sea-level rise, and soil salinization that erodes human settlements and infrastructure, driving many to seek alternative livelihoods, affecting the society and economy of Myanmar [21].

This research utilized spaceborne GNSS-R technology to analyze soil moisture sensitivity correlation between CYGNSS and SMAP in a regional study of Myanmar. Environmental impacts drive climate effect, and soil moisture changes were emphasized to study their correlations with the reflective signal, surface reflectivity, and soil moisture referenced data received by the SMAP passive reflectometry receiver. Figure 2 shows the different zonations according to the climatic condition of Myanmar.

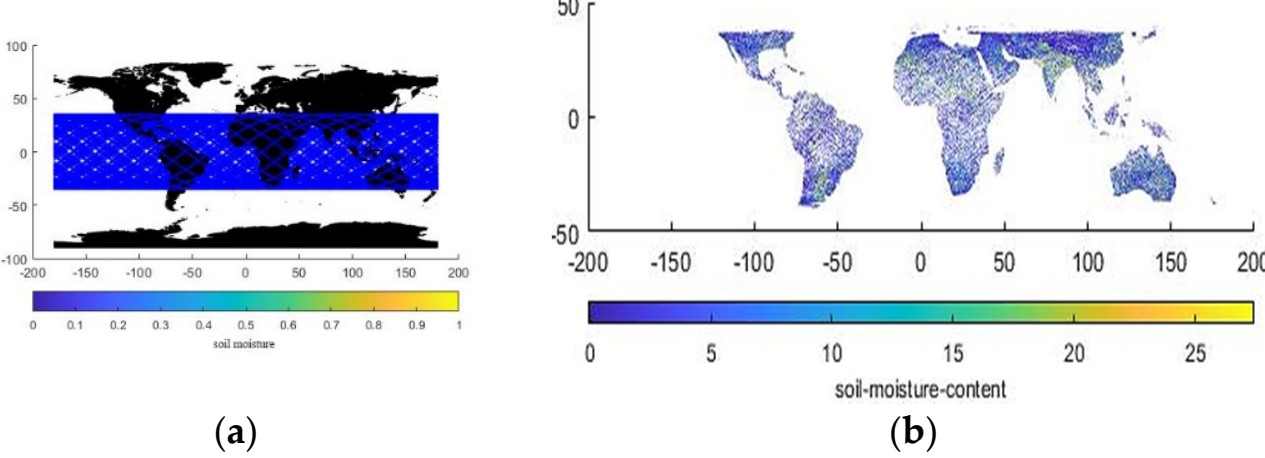

(**a**)      (**b**)

**Figure 2.** (**a**) CYGNSS L1 global data availability and (**b**) DDM SNR in August 2019.

### 3. Data Preparation

*3.1. CYGNSS L1 Data*

The CYGNSS level 1 bistatic radar cross-section of the Earth's surface provided a two-dimensional DDM grid of floating-point numbers. The CYGNSS satellite constellation uses the onboard Delay Doppler Mapping Instrument (DDMI) and delay Doppler maps (DDMs) calibrated into received power and bistatic radar cross-sections (brcs). Each of the CYGNSS satellites has a corresponding NetCDF (.nc format) data file provided each day. There are typically six–eight spacecrafts retrieving data under nominal conditions each day, but this can maximize to eight spacecrafts under particular circumstances. Higher than normal retrieval frequency is needed, for example, in tropical cyclone tracking cases. The Delay Doppler Map consists of the time delay function and Doppler frequency shift.

The GNSS-R-based soil moisture retrieval approach relies on an inversion of the DDM SNR bistatic radar equation to acquire the surface reflectivity. The surface reflectivity corrects the vegetation cover and surface roughness effects to obtain a Fresnel reflection coefficient. This coefficient is then related to soil moisture estimation by applying Fresnel reflection equations [22]:

$$\Gamma_{lr}(\theta) = |R_{lr}(\theta)|^2 x(z), \qquad (1)$$

where $R_{lr}$ is the Fresnel reflection coefficient and $x(z)$ is the probability density function of the surface height z. The Fresnel reflection coefficient $R_{lr}$ can be expressed as linearly polarization modes:

$$R_{lr} = R_{rl} = \frac{1}{2}(R_{vv} - R_{hh}), \qquad (2)$$

where $R_{vv}$ and $R_{hh}$ are Fresnel reflection coefficients for horizontal and vertical polarization.

The bistatic received power is for the coherent component where specular reflections are thoroughly dominant and robust; the bistatic received power is for the coherent component according to the Z-V model [23].

$$P_{RL}^{coh} = \left(\frac{\lambda}{4\pi}\right)^2 \frac{P_t G_t G_r}{(r_{st} + r_{sr})^2} \Gamma RL(\theta_i),$$
(3)

where $P_{RL}^{coh}$ is the coherently received SNR power, R and L stand for the right-hand circularly polarized (RHCP) GNSS transmit antenna and the LHCP downward-looking GNSS-R antenna, $\lambda$ is the free space wavelength, $P_t$ is the transmitted GNSS signals peak power, $G_t$ is the gain of the transmitter antenna, $G_r$ is the gain of the receiver antenna, $r_{st}$ is the distance between the GNSS transmitter and the specular reflection point, $r_{sr}$ is the distance between the GNSS-R receiver and the specular reflection point, and $\Gamma RL(\theta_i)$ denotes the specular reflectivity at a local incidence angle of $\theta_i$. $\lambda$ is the GPS wavelength (19 cm) and $\Gamma_{rl}$ is the reflectivity of the surface. The reflectivity of the surface $\Gamma_{rl}$ is affected by near-surface soil moisture, as soil moisture affects the surface dielectric constant, which affects surface reflectivity. Surface reflectivity is affected by soil surface roughness, small surface height irregularities on the scale of $\lambda$. Since this analysis was only concerned with changes in surface reflectivity, we omitted the '$4\pi$' and '$\lambda^2$' terms. Solving for reflectivity and making an additional correction for background noise (*N*), the surface reflectivity of $P_{r,eff}$ (in dB) [3] is then:

$$P_{r,eff} = 10\log\Gamma_{rl} \propto 10\log P_{rl}^c - 10\log N - 10\log G^r - 10\log G^t - 10\log P_r^t + 20\log(R_{ts} + R_{sr}),$$
(4)

The CYGNSS L1 DDM SNR was prepared for the data used. Data were collected in May, August, and October 2019 from the data product site, the Physical Oceanography Distributed Active Archive Centre (PODAAC), freely available at https://podaac.jpl.nasa.gov/dataset/CYGNSS_L1_V2.1, accessed on 9 April 2020 [24]. Figure 2a shows CYGNSS L1 data availability and Figure 2b shows the DDM SNR soil moisture contents prepared from the August 2019 monthly data. For Figure 3, we prepared DDM SNR and surface reflectivity $P_{r,eff}$. We collected data for May, August, and October 2019 to illustrate seasonal changes in the soil moisture study. We used the Equal-Area Scalable Earth (EASE 2.0) 9 km × 9 km grid cell in a global cylindrical projection with the same projection of SMAP. The EASE-Grid is based on a philosophy of digital mapping and gridding definitions developed at the University of Colorado at Boulder. It was intended to be a versatile scheme for global-scale gridded data users, especially those who use remotely sensed data, although it has been gaining popularity as a common gridding format for data from other sources as well [25].

Surface reflectivity $P_{r,eff}$ was higher in the delta and lower part of Myanmar in August because the surface was almost wet. In August 2019, the region had flash floods and triggering land instabilities, owing to tropical cyclones and weather depressions, particularly in the lower part of the country [26]. Most parts of the country have moist soil, and the surface temperature decreased because of the increasing surface reflectivity. Myanmar has a tropical monsoon climate, where 75–90% of the annual rainfall occurs in the monsoon season from June to September [27]. While monsoonal rains are essential for agriculture, supplying water for irrigation and alluvial sediments, these events can frequently lead to severe flooding, significantly impacting people, homes, and ecosystems.

### 3.2. SMAPL3 P Enhanced Data

3.2.1. Soil Moisture

The SMAP L3 Passive Reflectometry product was a daily global composite containing gridded data from 6:00 a.m. (descending) and 6:00 p.m. (ascending). SMAP passive radiometer-based soil moisture retrieval, ancillary data, and quality assessment flags on the global 9 km Equal-Area Scalable Earth (EASE 2.0) grid designed by the National Snow and Ice Data Center (NSIDC) [28]. The data were also collected for May, August, and October

2019 in the Myanmar regional study. SMAP L3 data are available on the NSIDC site and can be freely downloaded at https://nsidc.org/data/SPL3SMP_E/versions/3, accessed on 9 April 2020 [29].

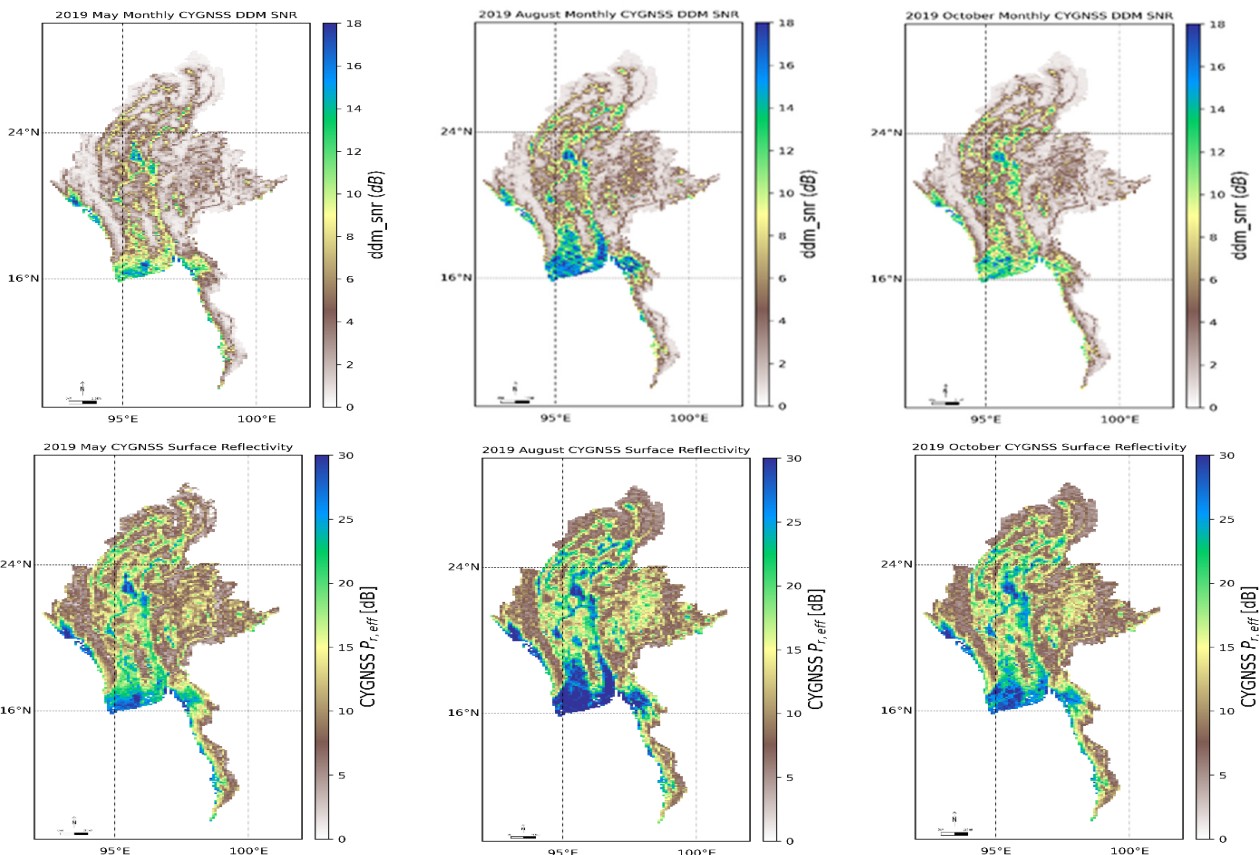

**Figure 3.** CYGNSS L1 DDM SNR and surface reflectivity of May, August, and October 2019.

SMAP used the L-band radiometer as a highly sensitive indicator of surface soil moisture. The radiometric brightness temperature represents emission determined mainly by the physical temperature and dielectric constant of the existing scene (soil moisture in the top ~5 cm). The sensitivity to soil moisture decreased significantly for surfaces with a vegetation water content (VWC) above ~5 kg m$^{-2}$. The SMAP radiometer instrument provided brightness temperature measurements with better than 1.3 K uncertainty (1-sigma), given that the brightness temperature difference across the dynamic range of surface soil moisture can be many tens of K (up to ~70 K and higher) [30]. Table 1 shows the SMAP Enhanced L3 Radiometer system overview of global daily 9 km EASE gridded soil moisture L3 data.

**Table 1.** Soil Moisture Active Passive (SMAP) system overview (source: O'Neill et al. 2019).

| Parameter(s) | Microwave > Brightness Temperature<br>Soils > Soil Moisture/Water Contents > Soil Moisture |
|---|---|
| Spatial Coverage | N: 85.044, S: −85.044, E: 180, W: −180 |
| Spatial Resolution | 9 km × 9 km |
| Temporal Coverage | 31 March 2015 to 27 August 2020 |
| Temporal Resolution | 1 Day |
| Data Format(s) | HDF5 |
| Platform(s) | SMAP |
| Sensor(s) | SMAP L-Band Radiometer |
| Version(s) | V3 |
| Data Contributor(s) | O'Neill, P. E., S. Chan, E. G. Njoku, T. Jackson, R. Bindlish, and J. Chaubell. |

SMAP-R received GPS L2 C-band (frequency = 1227.45 MHz, wavelength = 24.42 cm). The bistatic radar cross-section, which was the same equation used in CYGNSS calibrations but slightly modified to the SMAP-R characteristics, for the coherent assumptions [10] is as follows:

$$\sigma_{0coh} = \frac{(4\pi)^3 P_{coh}(\tau, f_d)\left(R_{rx_{sp}} + R_{tx_{sp}}\right)^2}{T_i^2 P_{tx} G_{tx} \lambda^2 G_{rx_{sp}} \; B \; (\tau, f_d)}, \tag{5}$$

where $\sigma_{0coh}$ is the bistatic radar cross-section of coherence surface, $P_{incoh}$, $P_{coh}$ is the power received from the incoherence and coherence surfaces, $P_{tx}$ is the GPS transmitted power, $G_{tx}$ is the GPS transmitter antenna gain, $\lambda$ is the GPS signal wavelength (at GPS-L2C is 24.42 cm), $R_{rx_{sp}}$ is the distance from the receiver to a particular surface pixel, $R_{tx_{sp}}$ is distance from the transmitter to a particular surface pixel, $T_i$ is the integration time, and $B\ (\tau, f_d)$ denotes the filtered scattering surfaces.

Coherent scattering is a surface dominated by any area containing mirror-like surfaces such as rivers, lakes, wetlands, flooded surfaces, or sea ice surfaces [10]. The Irrawaddy River begins in the upper highland area of Myanmar and flows to the lower parts of the delta area, then to the Adman Sea. Thus, both Myanmar and other parts of the Mekong region have the same experiences with flash water extending, flooding caused by storm surges, and other meteorological problems [31]. The following imagery was prepared with the enhanced soil moisture data from the SMAP L3 radiometer.

In Figure 4, the area is almost dry and has less soil moisture owing to the evapotranspiration process in May. In August, the regions were almost wet, and the soil was saturated in the monsoon season, then returned to a decrease at the beginning of the winter season in October.

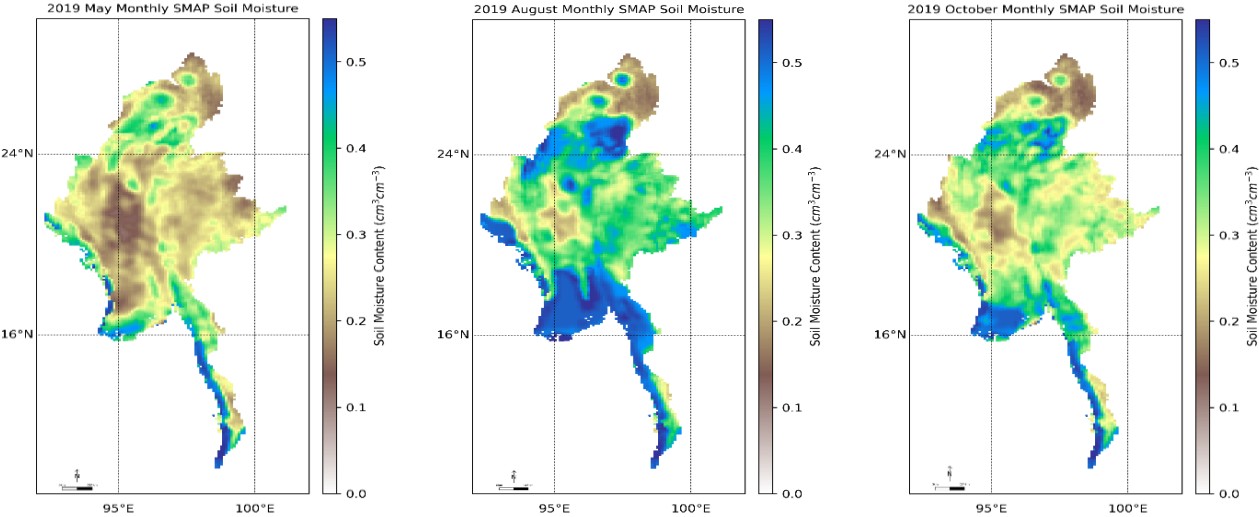

**Figure 4.** Monthly SM data received by the SMAP Passive Radiometer in May, August, and October 2019.

### 3.2.2. Surface Temperature

SMAP L3 surface temperature was obtained from L2 SM_P algorithms with ancillary datasets of AVHRR NDVI. The sufficient soil surface temperature was at 9 km grid spacing for the global daily composite data. According to surface temperature, the algorithms based on SMAP identified emissivity of the soil surface from brightness temperature (TB). This emissivity was used to sense soil moisture content. Figure 5 shows surface temperature monthly data prepared for May, August, and October 2019 in the Myanmar region. May was the hot and dry season in this region. The surface temperature was high owing to increased surface emissivity and brightness temperature (TB). The surface temperature dropped in the wet season of August and rose slightly again in October, turning into the winter (cold). In August, the country had moist soil and the surface temperature decreased

because of the surface reflectivity. The surface emissivity decreased, and the brightness temperature (TB) dropped.

**Figure 5.** Surface temperature received by the SMAP Radiometer in May, August, and October 2019.

### 3.2.3. Vegetation Opacity

According to scattering related the Wang's "tau omega" model (where $\tau$ is the optical thickness and $\omega$ the single scattering albedo of the vegetation canopy) [32], a daily global composite of the estimated vegetation opacity in the 9 km grid used the same "tau" parameter normalized by the cosine of the incidence angle:

$$\tau = \frac{b * VWC}{\cos \theta} \tag{6}$$

where b is a landcover-based parameter described in the SMAP Level 2/3 Passive Soil Moisture Product ATBD, VWC is vegetation water content in $kg/m^2$ derived from NDVI climatology, and $\theta$ is the incidence angle (= 40°) for SMAP.

According to the tau omega model, accounting for vegetation effects is essential for soil moisture retrieval. The presence of vegetation attenuates the soil emission. In addition, SMAP L3 reflectometry received soil moisture information influenced by brightness temperature and vegetation opacity [8].

If soil moisture content increased, surface brightness temperature increased accordingly. Surface roughness (h) was high, so soil moisture decreased because of the combined and incoherent scattering effect. Vegetation opacity was directly related to its water content, and it became higher in the monsoon season (wet) than in the dry and cold season [33]. Figure 6 shows that SMAP Radiometer vegetation opacity maps were prepared in May, August, and October 2019.

### 3.3. Processing Flows

In this research, we applied a GNSS-R bistatic radar cross-section technique based on the Z-V Model with the Cyclone Global Navigation Satellite System (CYGNSS) and SMAP observation dataset based on the tau omega ($\tau - \omega$) scattering model. Instead of a single instrument, CYGNSS comprises eight GNSS-R satellites in low Earth orbit around the tropical region. For this reason, we used the proposed technique with CYGNSS data because it provided more observation data and had a significant opportunity for soil moisture observation in the regional study.

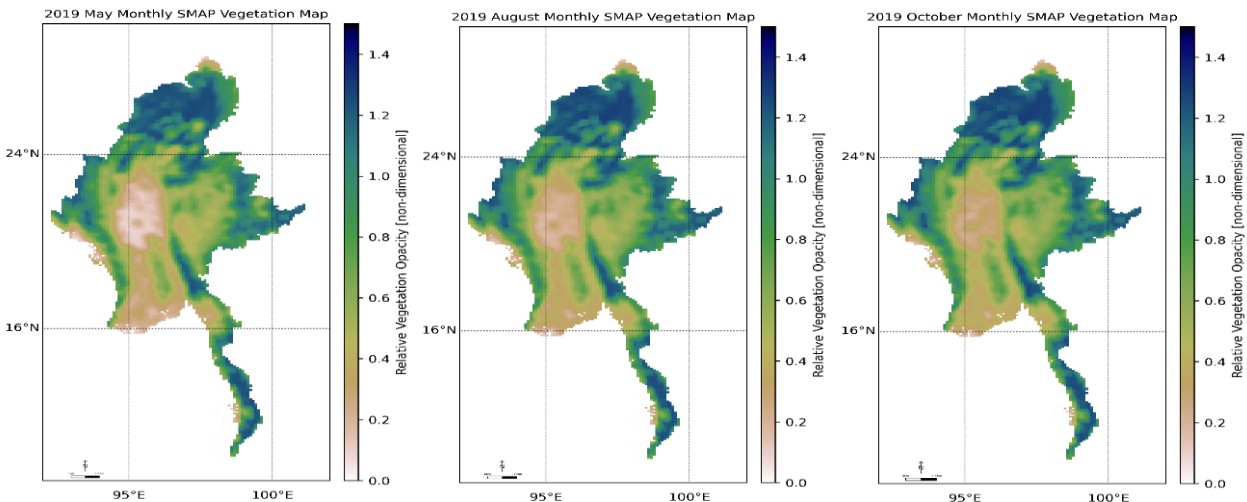

**Figure 6.** SMAP Radiometer vegetation opacity maps in May, August, and October 2019.

The theoretical footprint of a reflected CYGNSS signal is ~0.5 × 0.5 km (spatial resolution) in the case of a smooth surface (coherence) and a receiver at the altitude of CYGNSS, with slight dependence on the incidence angle [34]. The mean temporal resolution of CYGNSS is less than 12 h. SMAP reflectometry data were received at 36 km × 36 km (spatial) every day (temporal). In this research, we used the EASE 2.0 grid on the global 9 km × 9 km both for space and time collocation reference. For each grid cell, we calculated the daily average of the surface reflectivity $\Delta P_{r,eff}$ concerning the mean value for that grid cell for the entire period of interest (May, August, and October 2019). We calculated average mean soil moisture $\Delta SM$ in the same way: differences from the mean SM value for each grid cell in the seasonal observation period. We then compared the surface reflectivity $\Delta P_{r,eff}$ to the mean soil moisture $\Delta SM$ for each grid cell and calculated the correlation coefficient.

Figure 7 data processing flows diagram shows that spaceborne CYGNSS L1 initially received monthly data from May, August, and October 2019. Reflected received SNR power was extracted from the L1 radar cross-section of the Earth's surface products from all eight satellites of the CYGNSS mission. The received signal power was delivered as a function of the time delay (sampled according to the GNSS signal bandwidth) and Doppler frequency owing to platform and Earth motion (sampled according to the coherent integration time); the resulting Delay Doppler Map (DDM) represents the main observable.

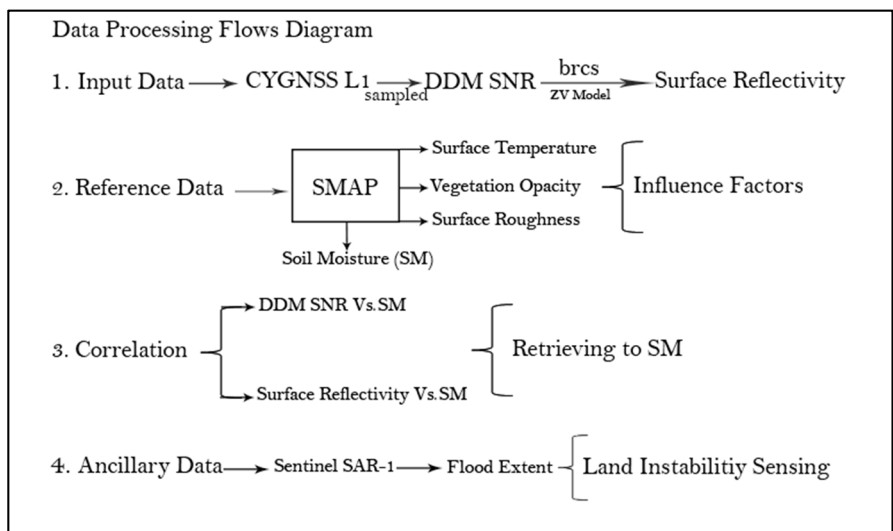

**Figure 7.** Processing flows of CYGNSS and SMAP for regional soil moisture sensing.

Then, DDM SNR converted to the surface reflectivity according to the bistatic scattering Z-V model. In the second step, we prepared soil moisture data and surface temperature, surface roughness, and vegetation opacity, which influenced the received signals of SMAP L3-enhanced radiometric data. Thirdly, the soil moisture retrieval process was carried out by CYGNSS DDM SNR and surface reflectivity was correlated with SMAP SM data. Finally, Sentinel SAR-1 data were validated to the land instability sensing for flood extended into monsoon season.

## 4. Results

This section discusses the correlation between CYGNSS DDM SNR and SMAP SM regarding the surface reflectivity response to soil moisture for diurnal changes in three different seasons of Myanmar.

### 4.1. Soil Moisture Correlation of CYGNSS DDM SNR and SMAP SM

In orbit, CYGNSS microsatellite observatories receive direct and reflected signals from GPS (Global Positioning System) satellites. The direct signal pinpoints CYGNSS observatory positions, velocity, and timing information, while the reflected signals respond to terrain characteristics [35]. Figure 8 shows that the CYGNSS receiver received reflected peak DDM SNR signal monthly in May, August, and October for seasonal changes (hot, wet, and cold) analysis. SMAP passive radiometer received monthly soil moisture data in the same collocation and time referenced.

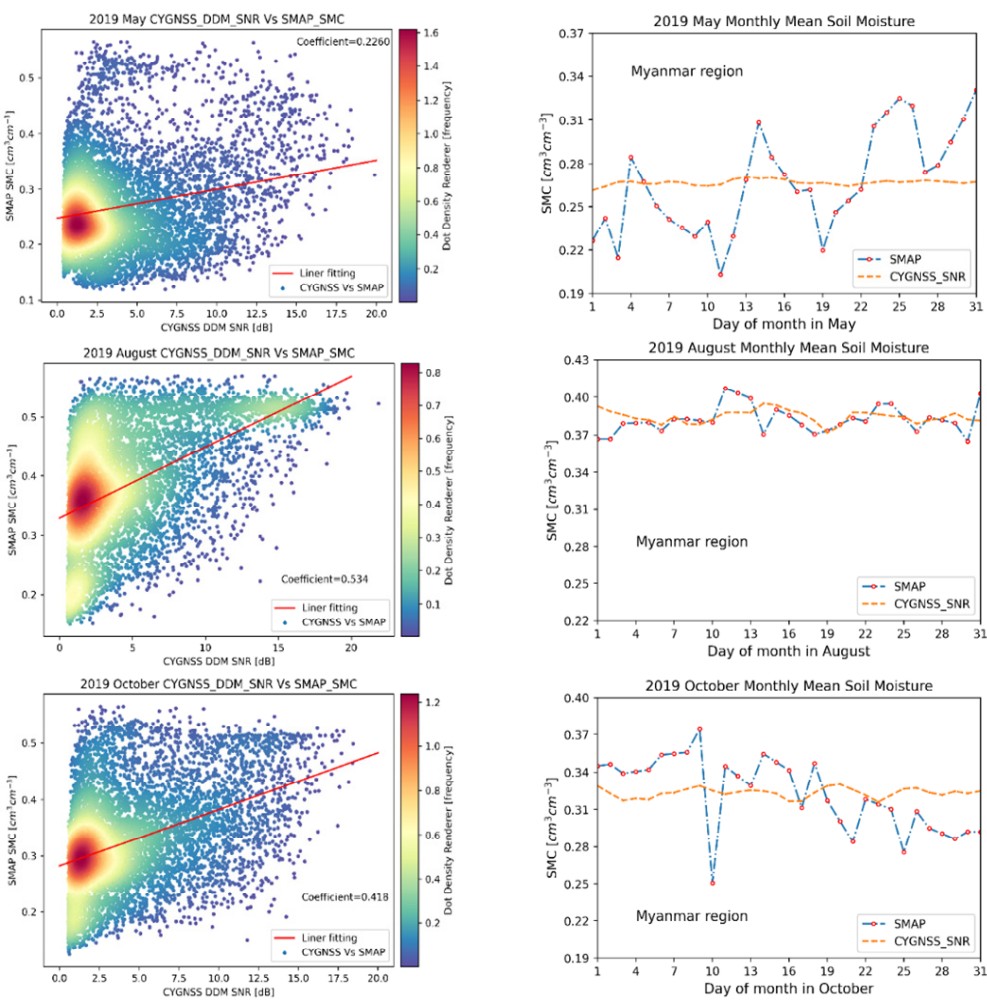

**Figure 8.** CYGNSS DDM and SMAP SM correlation and monthly mean soil moisture in May, August, and October 2019.

The DDM SNR received signal was sensitive to change according to correlation values. Soil moisture coefficient values were r = 0.2260 for May, r = 0.534 for August, and r = 0.418 for October. Significant changes and soil moisture retrieval were high in August 2019 owing to several weather conditions, i.e., flash floods from heavy rain and storm surges triggered in lower parts of Myanmar [36,37]. DDM SNR changed very significantly from dry to wet season. In the following Figure 8, the monthly mean soil moisture graph reflects that the obtained DDM SNR, and SMAP correlation was high and consistent in August. The surface was wet and greatly depended on other geophysical parameters such as vegetation canopy, brightness temperature, the dielectric constant of medium and terrain conditions, and scattering conditions for the coherent and incoherent phenomenon [11].

In Figure 9, Myanmar has three different seasonal changes, and the hot season ended in May. August was a wet (rainy) season. In October, the weather condition was temperate almost everywhere in the country. Thus, CYGNSS DDM SNR and SMAP were very sensitive to soil moisture over 3 months of different daily changes. the GNSS-R DDM SNR and soil moisture correlation was lower in the dry season, was higher in the wet season and saw a slight drop in the winter season. In August, the mean soil moisture value was highest at 0.40 $cm^3/cm^3$, which was a consistent and good agreement between CYGNSS and SMAP for spaceborne soil moisture sensing.

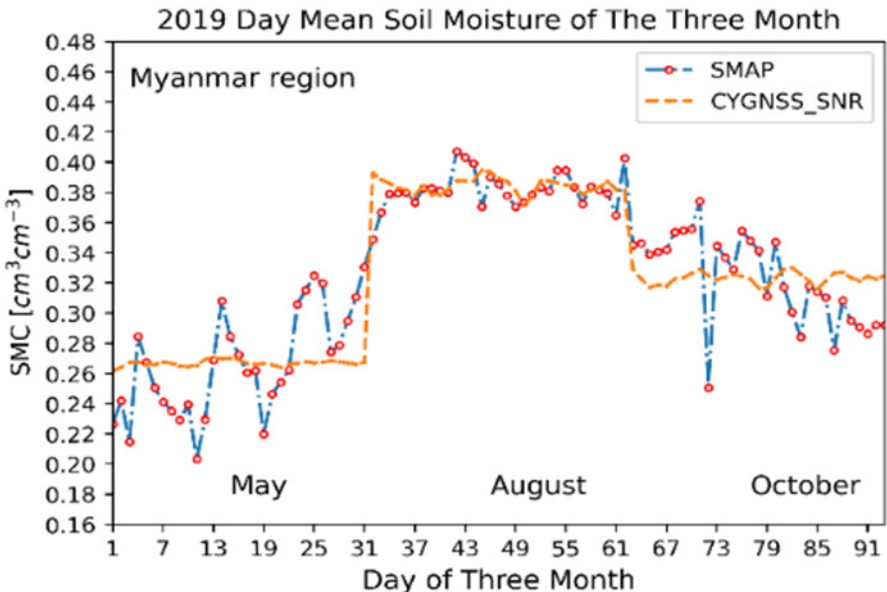

**Figure 9.** DDM SNR and SMAP diurnal mean SM in May, August, and October 2019.

### 4.2. Soil Moisture Correlation of CYGNSS Surface Reflectivity and SMAP SM

Figure 10 shows CYGNSS surface reflectivity and correlates it with the soil moisture data received by the SMAP L3 passive sensor. In the delta and lower part of Myanmar, severe flood conditions occurred from July to October owing to weather depression, causing a disaster crisis [38].

Correlation coefficient values of DDM SNR and its surface reflectivity were higher in August. CYGNSS surface reflectivity and SMAP SM correlation values were r = 0.2262 for May, r = 0.560 for August, and r = 0.461 for October. Myanmar had several disaster crises, such as flash floods due to weather depression. Flash floods caused landslides in some parts, and land instability occurred in the lower parts of Myanmar [39]. Monthly mean soil moisture was higher in August because the surface was almost always wet.

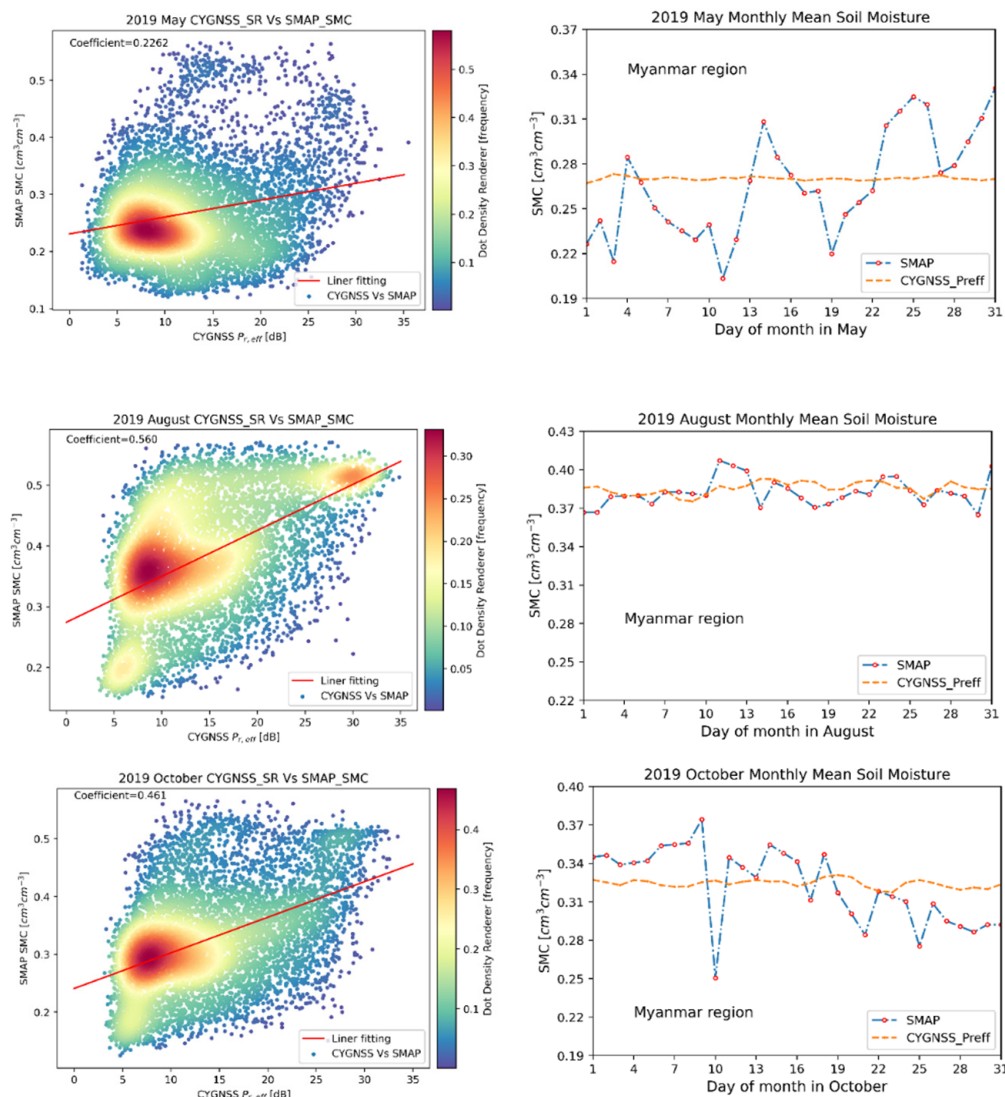

**Figure 10.** Surface Reflectivity $P_{r,eff}$ and SMAP correlation and monthly mean soil moisture in May, August, and October 2019.

The correlation calculation was completed using EASE 2.0 gridded to a 9 km × 9 km scale for CYGNSS and SMAP for the regional SM analysis. The correlation result was very different during the seasonal changes of May and August 2019. This meant a strong correlation (r = 0.560) between surface reflectivity $P_{r,eff}$ and SMAP SM in May and August.

Figure 11 shows daily mean soil moisture changes from May, August, and October 2019. As stated, earlier, the correlation coefficient value was higher in August, which meant daily soil moisture was significantly increased compared to other months. Myanmar had frequent meteorological depressions and tropical cyclones in the monsoon season until October. As a result, the surface area was more saturated, which influenced soil moisture content.

The research compared Chew et al.'s (2018) prediction for regional soil moisture observation on the medium vegetated area. However, they counted soil moisture variation from the surface reflectivity of CYGNSS DDM SNR data and SMAP; the soil moisture retrieving value was still low and not high enough to provide for heavily vegetated and rough terrain [3]. This research added the diurnal study with seasonal changes in soil moisture correlation to the regional analysis from NASA spaceborne observations of the CYGNSS and SMAP missions. There were consistent and linear relationships between

CYGNSS observations and SMAP SM retrievals, which were most easily quantified in the range where SM was highest.

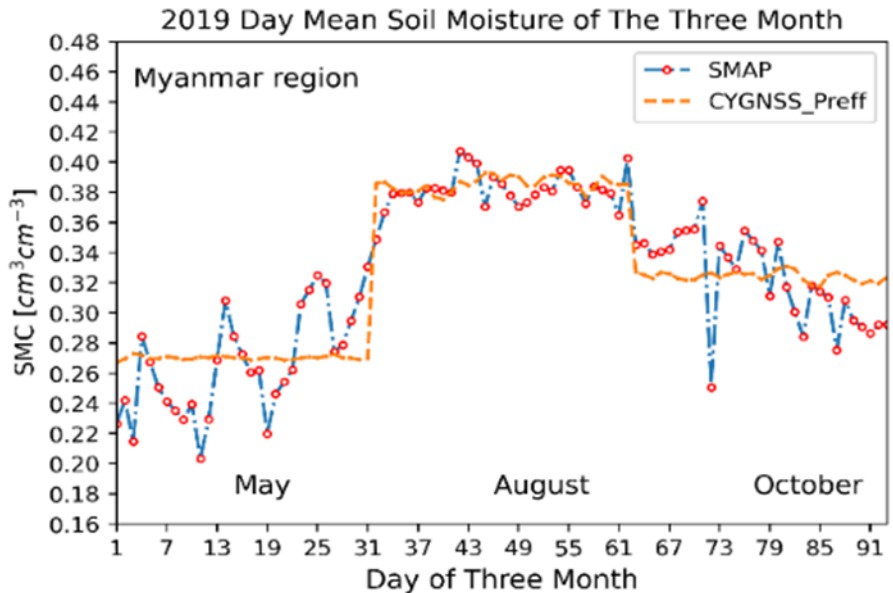

**Figure 11.** CYGNSS surface reflectivity and SMAP mean SM in May, August, and October 2019.

*4.3. Data Validation and Verification with SENTINEL SAR-1*

ESA, the Copernicus program, is being developed to launch the Sentinel-1 constellation of two polar-orbiting satellites. Sentinel-1 satellites operate all the time with C-band synthetic aperture radar imaging, enabling them to receive imagery regardless of the weather, day and night. Sentinel-1, a constellation of two (A and B) satellites with C-band SAR, provides backscatter (σ∘) observations at $5 \times 20$ m$^2$ resolution. Synthetic Aperture Radar (SAR) can penetrate clouds, regardless of weather conditions, and has day and night capability; many studies have focused on monitoring efficient and high-quality methods for surface water mapping.

Depending on the region, the SAR-based flood detection algorithms can detect floods with an accuracy ranging from 80 to 95%. Uddin et al. (2019) determined that Sentinel-1 SAR observation data have great potential in producing flood information with high accuracy and high spatial resolution over a 6-day interval, despite the predominance of severe weather conditions during flooding time in Bangladesh [40].

The Sentinel SAR-1 (A and B) constellation offers a 6-day exact repeat cycle. However, a single SENTINEL-1 satellite can trap the entire world once every 12 days. Accordingly, Sentinel SAR data cannot provide high temporal resolution for continuous flood detection purposes. Because of the dynamic nature of floods or flash floods, SAR images are not used operationally during floods [41]. Thus, no single standard dataset can satisfy all the conditions that fulfill the temporal evolution of flood inundation extent with reasonable accuracy [42].

In Sentinel SAR-1 imagery, each pixel value is directly related to the backscatter energy from the surface and terrain conditions. The Copernicus Sentinel SAR-1 images in Figure 12 were obtained after correcting the radiometric and geometric calibrations. In Figure 12b, Sentinel-1 SAR-1 data recorded flood-prone areas extending into the Mon State, in the southern part of Myanmar in August 2019. Monsoon rains affected Myanmar and other areas of Southeast Asia, submerging homes, displacing residents, and triggering landslides.

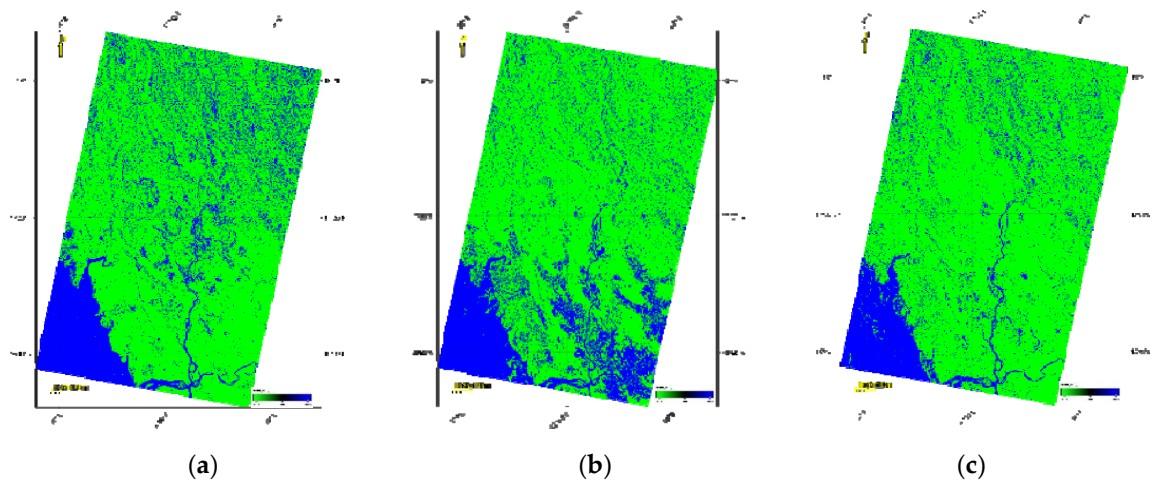

(a)                            (b)                            (c)

**Figure 12.** Sentinel SAR-1 images of the flood extension map; (**a**) May, (**b**) August, and (**c**) October 2019, Myanmar.

## 5. Discussions

In this analysis, CYGNSS reflectometry SNR data correlated with SMAP reflectometry soil moisture data. Moreover, CYGNSS reflectivity was compared to SMAP SM for monthly and daily observations in the study area. Image interpretation presented surface temperature, vegetation opacity, and surface roughness prepared from SMAP. Flood and land information analysis extracted data for validation and verification processes from Microwave Sentinel SAR-1. In the following Table 2, we compared correlation coefficient results of CYGNSS and SMAP SM.

**Table 2.** CYGNSS and SMAP SM correlation in the Myanmar region.

| Observed Date (2019) (Monthly) | DDM SNR vs. SMAP SM (Coefficient Value) | Surface Reflectivity vs. SMAP SM (Coefficient Value) |
|---|---|---|
| May | 0.2260 | 0.2262 |
| August | 0.534 | 0.560 |
| October | 0.418 | 0.461 |

A good agreement was observed between CYGNSS reflectometry and referenced soil moisture measurements from SMAP. The table shows the soil moisture correlation coefficient values for DDM SNR, and the surface reflectivity $\Delta P_{r,eff}$ was the highest in August (r = 0.534 and r = 0.560) and lowest in May (r = 0.2260 and r = 0.2262). It was expected that the region would be almost wet in August. The wet surface produced a stronger reflection than dry surfaces. The surface reflectivity $\Delta P_{r,eff}$ was mainly influenced by topography, surface water, and land cover, not only SM [6,43]. Moist surfaces have higher dielectric constants, which results in higher reflectivity than drier surfaces [13,44].

The soil moisture correlation results were generally consistent based on ancillary data interpreted from geophysical key parameters. CYGNSS received DDM SNR soil moisture sensitivity data on seasonal weather changes, land instabilities from flood intrusion, and tropical storm surge in the study area. The study demonstrated soil moisture from the CYGNSS mission on a regional scale for Myanmar, along with ancillary information about the overlying vegetation opacity and the surface roughness from SMAP passive reflectometry data. These results showed that the influences of other geophysical parameters, such as vegetation opacity, surface temperature, and surface roughness, were potential key issues for spaceborne GNSS-R sensing over land.

## 6. Conclusions

Satellite Navigation Technology principally pioneered its services to detect position, navigation, and time. After enhancement with GNSS reflectometry, it now helps the public in disaster mitigation, climate, hydro-meteorology forecasting, land information, and sea surface anomalies observation. Remote sensing reflectometry has emerged to detect abrupt changes in sea surface conditions, wind speed, and direction due to cyclone intrusions. GNSS reflectometry can also aid by using sea and ocean data and contemporarily apply them to land applications such as soil moisture, snow, time-series weather forecasting, and climatic observation. Up to the present, spaceborne observation for soil moisture sensing has been an uncertain process. The system, parameters, retrieving model, and algorithms remain a challenge to better understanding soil moisture observation. In addition, other geophysical parameters, such as surface roughness, vegetation biomass, and dielectric, are very complex issues.

The spaceborne NASA CYGNSS mission was initially intended to track tropical cyclones and well-observed land-based sensing. For this analysis, GNSS-R techniques retrieved several models such as the Z-V model of the bistatic cross-section, the tau omega ($\tau\omega$) scattering model, and the surface reflectivity model geophysical parameters. Although GNSS-R technology still has technical challenges, the eight CYGNSS satellites have improved system capabilities, and spatial and temporal resolution.

This research included a comprehensive analysis, including geophysical parameters such as reflectivity and surface roughness upon different incidence angles, vegetation optical depth (VOD), brightness temperature variations with coherent and incoherent scattering, and accurate information to deliver regional climatic and environmental concerns. A dielectric constant and soil moisture observation proved the data from CYGNSS and SMAP.

Finally, our research goal was to investigate spaceborne GNSS-R soil moisture observation and its influence by geophysical parameters. For a better understanding, soil-moisture-retrieving algorithms and all-inclusive models will need to guide uncertain spaceborne soil moisture sensing presenting challenging technical issues. New GNSS-R spaceborne missions will also be comparatively challenging to an existing system. As a result, multidisciplinary science can meet human needs in one way or another to sustain the planet inhabited by human beings.

**Author Contributions:** A.L. and D.Y. devised the initial idea for this research; A.L., X.H. and B.Z. (Bo Zhang) designed the algorithm. A.L. and B.Z. (Baoyin Zhang) contributed to the analysis of the results. C.S.S. provided the data. A.L., D.Y. and X.H. finalized the writing, review, and editing of the paper. All authors have read and agreed to the published version of the manuscript.

**Funding:** This research was funded by Taishan Industry Leading Talents of Shandong Province, China (Grant No. lzbz2016190).

**Institutional Review Board Statement:** Not applicable.

**Informed Consent Statement:** Not applicable.

**Data Availability Statement:** Data availability excluded.

**Acknowledgments:** We would like to express our gratitude for the spaceborne data provided by institutions and space agencies such as Surrey Satellite Technology, ESA, NASA, and the National Oceanic Atmospheric Administration (NOAA). Additionally, we acknowledge the use of Sentinel SAR 1 data from the Copernicus Services (https://scihub.copernicus.eu/, accessed on 9 April 2020), which is part of the European Space Agency (ESA).

**Conflicts of Interest:** The authors declare no conflict of interest.

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
