# Peer review of "Opportunity for GNSS Reflectometry in Sensing the Regional Climate and Soil Moisture Instabilities in Myanmar"

_climate, doi:10.3390/cli9120175_

Round 1

Reviewer 1 Report

Review of the manuscript entitled "GNSS Reflectometry Opportunity on the Regional Climate and Soil Moisture Instabilities Sensing in Myanmar"

General Comments: In my opinion, this research is valuable and I find some merits in these manuscripts. The manuscript has a good potential to be published in climate journal (MDPI).  However, I have concerns that the authors need to address carefully. The current need significant improvement based on the following detailed comments. I recommend a major revision.

Detailed comments:

Title: The title is appropriate

Abstract: Overall, the abstract the language flow need slight editing to improve flow. I suggest minor language editing to improve this section.

Keywords: The authors provided seven keywords. The keywords are appropriate and would help the article search results in the future or increase the article's visibility to a large audience.

Introduction: In this section, the authors provide some description of the topic. the introduction cited specific recent references in a broader context of the international literature available on the topic. However, I have some concerns regarding the flow of this section. The goal is not clearly stated and the structure of the section needs to be improved. In most instances, reading is not smooth and most of the sentence structure is too long. Readers have to read at least twice to follow the manuscript. Ivsuggest the authors split sentences for many of the long sentences. Last but not least, I suggest language editing to improve this section.

Methodology: Overall, the research procedures and techniques used are standard for this scientific research and are reproducible. In many instances, it is not clear to differentiate what the authors did and what they relied on based on previous studies. For example, L1, L2 and L3 datasets needs different levels of processing by authors. Authors to need to clearly state what data processing they did at each level before using a specific data. In addition, the authors fail to cite statements made in this section. I strongly recommend statements be duly cited. A statement to make state the limitation(s) in this study may be useful. Moreover, I think language editing is needed to improve the overall of the manuscript.

Minor comments

Line 154: provide source for Figure 3

Line 158-160: provide citation for this statement

Line 162: provide citation for this statement

Line 165-169: provide citation for this statement

Line 182: provide source for this figure 5.

Line 193-198: provide references for this paragraph

Line 201-202: “In figure 6, the area has almost dry and less soil moisture as the evapotranspiration 201 process in May.” Not clear!! Do you mean to say “In figure 6, the area has almost dry and less soil moisture due to the evapotranspiration process in May”?

Line 218: 3.2.3. Vegetation Opacity- provide citation for statements in 222. For 223-226 needs citations

Line 237: Provide legend for figure 9. The shapes need clear legend explain what each different shape represent or mean. Disconnection in linking the shapes. Same for subsection 228

Results: The results are clear, well presented. Overall, well complemented with enough figures that help to visualize the results. However, no comparison is made with previous studies. I think language editing is needed to improve the overall of the manuscript.

Minor comments

Line 256-259: provide citation to support this statement.

Line 252-254:  provide citation for this statement

Line 278-279: Provide citation to support this statement

Line 283-287: provide citation to support claim

Discussion: This section looks like a summary or conclusion than a discussion. I did not see any significant discussion of the results. Out of 29 references used in this manuscript, only one citation was provided in this section. I suggest this section needs to be rewritten. The connection of the study to extreme events such as floods etc. is vague and authors need to find a way to connect this section.

Conclusion: This section is well written and needs only language editing

 References

References are current and relevant for the study. However, few reference citations are incomplete

Minor comments

Line 443: Reference citation is incomplete

Line 458: Citation is incomplete

Recommendation

Accept with major revision

Reviewer 2 Report

  • In the present manuscript, the authors used GNSS R to assess soil moisture correlation with CYGNSS and SMAP in Myanmar. it is not clearly written what additional knowledge that the authors are proposing to the research community other than using existing methods to calculate the correlations between various datasets. The authors have stated the preliminaries to explain the concepts used in the paper. However, their contribution is not clear.

  • The introduction part needs enhancement by providing some of the most recent studies in this area. I suggest adding the following literature and providing some discussions about the differences and similarities of the results achieved by this study and findings of these articles.

Humphrey, V., Berg, A., Ciais, P., Gentine, P., Jung, M., Reichstein, M., ... & Frankenberg, C. (2021). Soil moisture–atmosphere feedback dominates land carbon uptake variability. Nature592(7852), 65-69.

Gavahi, K., Abbaszadeh, P., Moradkhani, H., Zhan, X., & Hain, C. (2020). Multivariate Assimilation of Remotely Sensed Soil Moisture and Evapotranspiration for Drought Monitoring. Journal of Hydrometeorology, 21(10), 2293-2308.

  • The language improvements. The text suffers from many grammatical and structural errors.

  • It is not clear at what spatial resolution the SMAP data set was used. And how the authors accounted for the spatial and temporal mismatches between the datasets used.

  • Figure 13 caption mentions “CYGNSS surface reflectivity Vs. SMAP Mean SM correlation changes” but no correlation coefficient can be seen in this figure. I suggest adding the correlation coefficient value to the figure.

  • The discussion section is too short. The authors need to elaborate more on how the obtained results contribute to enhance the community knowledge of this area

  • In the conclusion section, the limitations of this study and suggested improvements of this work should be highlighted.

Round 2

Reviewer 1 Report

The authors have acknowledge and addressed comments raised. A statement stating some challenges or limitation(s) in this study may be useful. I recommend the manuscript for publication.

Reviewer 2 Report

The authors have fully addressed my comments and I suggest its publication in Climate.